# Estimating the impact of alternative programmatic cotrimoxazole strategies on mortality among children born to mothers with HIV: A modelling study

**Shrey Mathur**[1]*, **Melanie Smuk**[1], **Ceri Evans**[1,2,3], **Catherine J. Wedderburn**[4,5], **Diana M. Gibb**[4], **Martina Penazzato**[6], **Andrew J. Prendergast**[1,2]

**1** Blizard Institute, Queen Mary University of London, London, United Kingdom, **2** Zvitambo Institute for Maternal and Child Health Research, Harare, Zimbabwe, **3** Department of Clinical Infection, Microbiology and Immunology, University of Liverpool, Liverpool, United Kingdom, **4** Medical Research Council Clinical Trials Unit at University College London, London, United Kingdom, **5** Department of Paediatrics and Child Health and Neuroscience Institute, University of Cape Town, Cape Town, South Africa, **6** Department of Research for Health, Science Division, World Health Organization, Geneva, Switzerland

* s.mathur@qmul.ac.uk

**Data Availability Statement:** The data underlying the results presented in the study are available from https://osf.io/8kjgp/.

## Abstract

### Background

World Health Organization (WHO) guidelines recommend cotrimoxazole prophylaxis for children who are HIV-exposed until infection is excluded and vertical transmission risk has ended. While cotrimoxazole has benefits for children with HIV, there is no mortality benefit for children who are HIV-exposed but uninfected, prompting a review of global guidelines. Here, we model the potential impact of alternative cotrimoxazole strategies on mortality in children who are HIV-exposed.

### Methods and findings

Using a deterministic compartmental model, we estimated mortality in children who are HIV-exposed from 6 weeks to 2 years of age in 4 high-burden countries: Côte d'Ivoire, Mozambique, Uganda, and Zimbabwe. Vertical transmission rates, testing rates, and antiretroviral therapy (ART) uptake were derived from UNAIDS data, trial evidence, and meta-analyses. We explored 6 programmatic strategies: maintaining current recommendations; shorter cotrimoxazole provision for 3, 6, 9, or 12 months; and starting cotrimoxazole only for children diagnosed with HIV.

Modelled alternatives to the current strategy increased mortality to varying degrees; countries with high vertical transmission had the greatest mortality. Compared to current recommendations, starting cotrimoxazole only after a positive HIV test had the greatest predicted increase in mortality: Mozambique (961 excess annual deaths; excess mortality 339 per 100,000 HIV-exposed children; risk ratio (RR) 1.06), Uganda (491; 221; RR 1.04), Zimbabwe (352; 260; RR 1.05), and Côte d'Ivoire (125; 322; RR 1.06). Similar effects were observed for 3-, 6-, 9-, and 12-month strategies. Increased mortality persisted but was

**Funding:** SM is supported by a National Institute for Health and Care Research (NIHR) Academic Clinical Fellowship (https://www.nihr.ac.uk/, award ref: ACF-2020-19-502). AJP is funded by the Wellcome Trust (https://wellcome.org/, grant number: 108065/Z/15/Z). The funders had no role in study design, data collection and analysis, decision to publish, or preparation of the manuscript.

**Competing interests:** The authors have declared that no competing interests exist.

**Abbreviations:** ART, antiretroviral therapy; EID, early infant diagnosis; HEU, HIV-exposed uninfected; PMTCT, prevention of mother-to-child transmission; RR, risk ratio; WHO, World Health Organization.

attenuated when modelling lower cotrimoxazole uptake, smaller mortality benefits, higher testing coverage, and lower vertical transmission rates. The study is limited by uncertain estimates of cotrimoxazole coverage in programmatic settings; an inability to model increases in mortality arising from antimicrobial resistance due to limited surveillance data in sub-Saharan Africa; and lack of a formal health economic analysis.

## Conclusions

Changing current guidelines from universal cotrimoxazole provision for children who are HIV-exposed increased predicted mortality across the 4 modelled high-burden countries, depending on test-to-treat cascade coverage and vertical transmission rates. These findings can help inform policymaker deliberations on cotrimoxazole strategies, recognising that the risks and benefits differ across settings.

## Author summary

### Why was this study done?

- Cotrimoxazole prophylaxis is recommended in World Health Organization (WHO) guidelines for all children born to mothers with HIV until HIV infection has been excluded by an age-appropriate HIV test to establish the final diagnosis after complete cessation of breastfeeding.

- Though there is a proven mortality benefit for children who acquire HIV, recent trial evidence has shown that cotrimoxazole does not reduce mortality for majority of children who are HIV-exposed uninfected (HEU), which has led to countries considering changing their guidelines.

- In many resource-limited settings, however, it is difficult to reliably distinguish children with HIV from children who are HEU, due to incomplete coverage of early infant diagnosis (EID) of HIV.

- There is a need to model to what extent alternative cotrimoxazole strategies, which either do not provide universal cotrimoxazole for all infants who are HIV-exposed, or provide it for a shorter duration, would be predicted to increase mortality in different settings among infants who acquire HIV but are undiagnosed.

### What did the researchers do and find?

- This study uses mathematical modelling based on epidemiological data from 4 high-burden settings (Côte d'Ivoire, Mozambique, Uganda, and Zimbabwe) to estimate the effect on mortality of alternative programmatic cotrimoxazole strategies.

- The model incorporates the HIV status of the infant, perinatal and postnatal transmission rates, testing rates, and mortality benefits from trial evidence for cotrimoxazole and antiretroviral therapy (ART) across 6 different programmatic strategies: maintaining current recommendations; reducing the duration of cotrimoxazole provision to 3, 6, 9, or 12 months; or starting cotrimoxazole only once a child tests positive for HIV.

- We demonstrate that changing the current strategy is predicted to increase mortality in all 4 settings, with the greatest increase in mortality in countries with the highest vertical transmission rates.

- Increased predicted mortality persisted in sensitivity analyses considering conservative model estimates, although cotrimoxazole had fewer predicted benefits when vertical transmission rates were lowered, testing coverage improved or uptake of cotrimoxazole was reduced.

## What do these findings mean?

- Changing the current strategy of cotrimoxazole provision for all children born to mothers with HIV is estimated to increase mortality in these 4 high-burden settings to varying degrees as countries continue to scale up prevention of mother-to-child transmission (PMTCT) of HIV and EID coverage.

- Cotrimoxazole continues to provide important protection to children who acquire HIV and are missed by gaps in the test-to-treatment cascade, but does not replace the importance of timely testing and treatment.

- Our study is limited by lack of cost-effectiveness analysis, lack of data on cotrimoxazole uptake, and limited antimicrobial resistance surveillance data in sub-Saharan Africa.

- Policymakers need to weigh the risks and benefits of cotrimoxazole prophylaxis through any change to current recommendations, noting that these differ across settings: where lower vertical transmission rates and improved testing and treatment uptake occurs, the estimated mortality benefits of cotrimoxazole are attenuated.

## Introduction

Vertical transmission of HIV occurs during pregnancy, labour, delivery, and breastfeeding. Despite substantial reductions in vertical transmission globally due to increased uptake of antiretroviral therapy (ART) among pregnant and breastfeeding women, transmission rates remain at 9% for Eastern and Southern Africa and 21% for Western and Central Africa [1]. Poor uptake of prevention of mother-to-child transmission (PMTCT) interventions arises due to late antenatal booking, low rates of HIV testing and ART initiation, and disengagement with services during pregnancy or breastfeeding [2]. Infants acquiring HIV have rapid disease progression, with over 50% mortality by 2 years of age without infant ART [3]. Given ongoing high transmission rates, limited PMTCT uptake, and rapid progression in undiagnosed children, programmatic strategies are needed to reduce morbidity and mortality among children born to mothers with HIV, until elimination of vertical transmission is achieved.

Cotrimoxazole is an inexpensive, well-tolerated, broad-spectrum antibiotic, which is recommended in World Health Organization (WHO) guidelines for all infants born to mothers with HIV from age 4 to 6 weeks until the end of HIV-exposure—typically defined as the end of breastfeeding, 18 months of age, or conclusive determination of being HIV-free [4,5]. There is strong evidence of mortality benefit from receiving cotrimoxazole among children with HIV. The CHAP trial demonstrated a 43% reduction in mortality in ART-naïve children, and the ARROW trial showed that benefits of cotrimoxazole persist despite immune reconstitution on

long-term ART [6,7]. Conversely, there is no evidence of mortality benefit from cotrimoxazole among children who are HIV-exposed but uninfected (HEU) [8,9]. Furthermore, there are concerns about antibiotic resistance and microbiome dysbiosis with cotrimoxazole use [10]. Given the distinct cotrimoxazole strategy required in each group, there is a critical need to identify and distinguish children with HIV from children who are HEU across different periods of transmission.

HIV testing for children has improved substantially over the past 2 decades. However, only 62% of infants globally currently receive the recommended virological test within 2 months of birth [1]. Overall, 40% of children with HIV therefore remain undiagnosed [11]. Even where infants are tested, challenges remain across the testing-to-treatment cascade. These include lengthy turnaround times for results, lack of integration with child health services, loss to follow-up, and delayed ART initiation [12,13]. Vertical transmission through prolonged breastfeeding now accounts for over half of transmissions in some settings and necessitates longitudinal testing [2]. WHO guidelines encourage testing of HIV-exposed children at 9 months of age and after complete cessation of breastfeeding [4].

A recent systematic review highlighted the lack of mortality benefit from cotrimoxazole for children who are HEU [14]. However, in settings with insufficient PMTCT and inadequate early infant diagnosis (EID) programmes, there is a substantial population of undiagnosed infants with HIV, who may benefit from cotrimoxazole prophylaxis to reduce mortality. In other settings, where PMTCT uptake and EID coverage are high and the majority of infants are HEU, the risk-benefit balance of cotrimoxazole may differ. Therefore, given these competing concerns, the optimal duration of cotrimoxazole prophylaxis needs to be determined. Here, we model the mortality impact of alternative cotrimoxazole strategies in 4 high-burden settings.

## Methods

### Model structure

We constructed a deterministic compartmental model with a decision tree to compare alternative cotrimoxazole prophylaxis strategies for children who are HIV-exposed. The model aimed to mimic epidemics in 4 high-burden countries: Zimbabwe (ZWE), Côte d'Ivoire (CIV), Mozambique (MOZ), and Uganda (UGA). These countries represent differing sub-Saharan African contexts and have sufficient information to enable estimates for the model parameters. The model structure and all estimates or assumptions applied are shown in Fig 1 and Table A in S1 Text. The study did not have a prospective protocol. This study is reported as per the transparent reporting of a multivariable prediction model for individual prognosis or diagnosis (TRIPOD) guideline (S1 Checklist).

The model estimates mortality from 6 weeks of age, when cotrimoxazole initiation is recommended, to 24 months of age. Testing time points were assigned at 6 weeks (defined as EID), at 9 months, and at 18 months (defined as end of breastfeeding, EoB). At 6 weeks, infants are categorised according to whether they have acquired perinatal HIV and whether they undergo EID. Children who are HIV-free at 6 weeks may acquire HIV through breastfeeding and may or may not receive additional HIV testing at 9 months and end of breastfeeding. Cotrimoxazole was provided based on programmatic strategy (Fig 1B). If the child was diagnosed with HIV, cotrimoxazole was started in conjunction with ART. ART was initiated only after a positive HIV test and was not initiated after negative or missed tests (i.e., for undiagnosed cases). Disease progression was incorporated through survival probabilities for children with HIV and children who are HEU, extracted from trial data and natural history studies [3,15]. Mortality occurs at each stage of the model and depends on current age, HIV status, ART use, and cotrimoxazole use. We defined model parameters for HIV transmission,

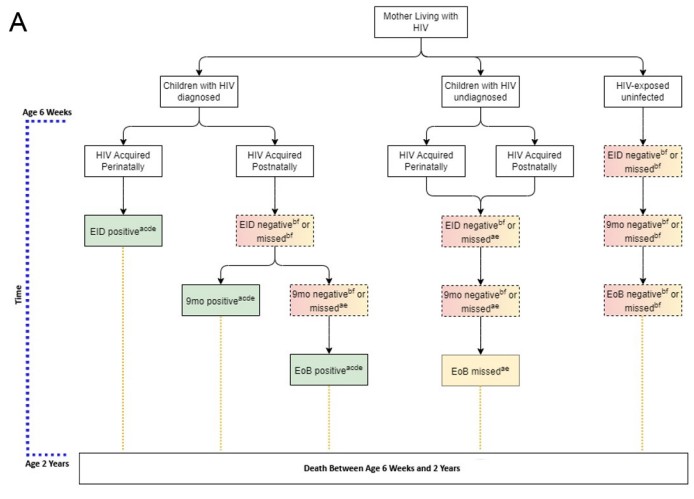

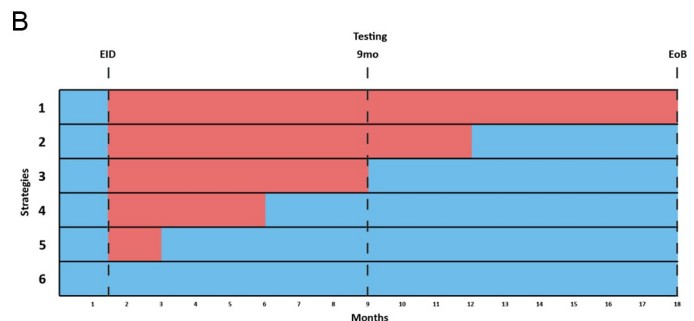

**Fig 1. Schematic illustration of model structure (scenarios and strategies). Panel A (Scenarios):** Scenarios considered were (1) children with HIV (diagnosed), (2) children with HIV (undiagnosed), or (3) HEU. Superscripts denote applied mortality estimates: (a) age-specific baseline mortality estimates for infants with HIV and no ART, based on Newell and colleagues [3]. (b) Age-specific baseline mortality estimates for HEU infants based on Arikawa and colleagues [15], (c) 77% uptake of ART following positive HIV test based on Luo and colleagues [17], (d) 76% mortality reduction from ART based on CHER [18], (e) 43% mortality reduction from CTX in a child with HIV based on CHAP [7], (f) 0% mortality reduction from CTX in an HEU child based on Botswana and South Africa trials [8,9]. **Panel B (Strategies):** Illustration of considered programmatic strategies. Pink represents period of programmatic CTX provision. Blue represents period of risk of HIV transmission due to breastfeeding (18 months). Strategies illustrated are (1) current WHO strategy (2) initiating CTX at 6 weeks and stopping at 12 months, (3) initiating CTX at 6 weeks and stopping at 9 months, (4) initiating CTX at 6 weeks and stopping at 6 months, (5) initiating CTX at 6 weeks and stopping at 3 months, and (6) initiating CTX prophylaxis only once a positive HIV test result is confirmed and the child is started on ART. CTX, cotrimoxazole; ART, antiretroviral therapy; EID, early infant diagnosis at 6 weeks; 9mo, 9-month HIV test; EoB, end of breastfeeding; HEU, HIV-exposed uninfected.

coverage of EID and ART uptake using epidemiological data, as outlined below, while the effects of cotrimoxazole and ART on mortality were derived from trials and meta-analyses (Table A in S1 Text) [2,3,15–18].

Compartments were created to compare strategies of receiving cotrimoxazole: (1) from age 6 weeks to end of breastfeeding (current WHO strategy; base case); (2) from 6 weeks to 12 months; (3) from 6 weeks to 9 months; (4) from 6 weeks to 6 months; (5) from 6 weeks to 3 months; and (6) only once a positive HIV test result is confirmed. In all strategies, we assume that infants testing positive for HIV will start cotrimoxazole.

The primary outcome of deaths among children who are HIV-exposed from age 6 weeks to 2 years is expressed as (1) percentage mortality; (2) risk ratio (RR) compared to the current strategy; (3) excess deaths per year relative to the current strategy; and (4) excess annual deaths

per 100,000 compared to the current strategy. The number of infants who are HIV-exposed in each country was calculated based on the number of live births and proportion of mothers living with HIV [19].

## Data sources and parameters

We included parameters to reflect child HIV status based on UNAIDS perinatal and postnatal transmission rates. The probability of an HIV test yielding a positive result was based on age-specific transmission rates, obtained from country-specific UNAIDS estimates from the most recent year with available data for all countries, including six-week transmission rates (MOZ 6%, ZWE 5%, CIV 4%, and UGA 3%) [2]. For postnatal transmission, we calculated the cumulative probability of acquiring HIV by age 9 months (MOZ 3.4%, ZWE 1.7%, CIV 1.7%, UGA 1.3%), and by the end of breastfeeding (MOZ 4.1%, ZWE 2.0%, CIV 2.1%, UGA 1.6%), based on derived weekly transmission rates from country-specific final vertical transmission rates and breastfeeding duration, making the assumption that postnatal transmission rates were consistent over time [16].

EID rates at 6 weeks were derived from country-specific UNAIDS estimates (MOZ 82.9%, ZWE 75.9%, UGA 66.2%, CIV 60.8%) [16]. Since data on subsequent testing are not available, we assumed that 80% of children undergoing EID would have a 9-month test and 50% would have an EoB test, as these children would likely be engaged in the test-to-treatment cascade. For children not undergoing EID at 6 weeks, we assumed the probability of subsequent testing was much lower (10% for 9-month test, 30% for EoB), as these children were likely less engaged with services. We derived the proportion of children starting ART after a positive HIV test result based on point-of-care and standard-of-care testing coverage and corresponding ART uptake from a recent meta-analysis [17].

## Scenarios

To estimate mortality, we considered several scenarios: (1) children with HIV diagnosed by EID; (2) children with HIV not yet diagnosed (due to missed 6-week, 9-month, and/or EoB tests) or with postnatal acquisition after an earlier negative test; and (3) children who are HIV-exposed and uninfected.

In the first scenario, children with known positive HIV status after EID started cotrimoxazole and 77% received ART based on a recent meta-analysis [17]. We applied a 43% mortality reduction following cotrimoxazole initiation based on results of the CHAP trial [7], and a 76% mortality reduction following ART initiation based on the CHER trial, to underlying mortality estimates [3,18].

In the second scenario, children had undiagnosed HIV and did not therefore receive ART. We based the survival probability on a pooled analysis of natural history studies conducted prior to the availability of ART [3] and applied a 43% mortality reduction due to cotrimoxazole alone. With the same rationale as in the first scenario, for 77% of children diagnosed with HIV (by 9 month or EoB testing), we applied mortality reduction of 76% from ART and 43% from cotrimoxazole.

In the third scenario, we derived mortality estimates for children who are HEU from an individual pooled analysis [15] and applied no mortality reduction due to cotrimoxazole, based on trial data [8,9].

## Sensitivity analyses

To explore uncertainties in the model assumptions, we conducted sensitivity analyses by adjusting estimates of cotrimoxazole uptake, mortality reduction from cotrimoxazole, vertical

transmission rates (perinatal and postnatal), and EID testing rates. For cotrimoxazole, we considered the effect of lower uptake (40%, 60%, and 80% of HIV-exposed children starting cotrimoxazole at age 6 weeks) compared to 100% uptake in the base model. For mortality effects of cotrimoxazole, we altered the base case assumption of 43% mortality reduction, derived from the CHAP trial, which compared cotrimoxazole and placebo among hospitalised children with HIV in Zambia. Since CHAP included very few infants, meaning mortality reductions are uncertain in children <12 months old, we used conservative estimates of 15%, 20%, 25%, 30%, 35%, 40% mortality reduction from cotrimoxazole to test a range of assumptions. To consider future progress in the prevention of MTCT, we reduced perinatal MTCT in 1% decrements to 0% and, separately, reduced postnatal MTCT rates in 0.5% decrements to 0%. Finally, we increased EID coverage in 10% increments, from current country rates up to 100%, to estimate the mortality benefits of cotrimoxazole with improved testing coverage over time. In a final sensitivity analysis, we combined the variables above to create a conservative estimate of effects, assuming low levels of cotrimoxazole uptake (40%), low ascribed mortality reduction from cotrimoxazole (15%), low perinatal (1%) and postnatal (1%) vertical transmission, and optimal EID coverage (100%).

### Data access and ethics

The model was constructed in Microsoft Excel, is freely available at https://osf.io/8kjgp/, and can be adapted for other settings. No ethical approval was sought for this modelling study.

## Results

Modelling the current WHO strategy, predicted mortality was 5.58% for Zimbabwe, 5.59% for Côte d'Ivoire, 5.78% for Mozambique, and 5.39% for Uganda. Limiting cotrimoxazole exposure from 6 weeks to 3 months of age, resulted in higher predicted mortality in each setting compared to the current strategy: Zimbabwe (increase in predicted mortality from 5.58% to 5.83%, RR 1.04, predicted excess mortality rate 246 deaths per 100,000 infants HIV-exposed, or 334 excess annual deaths), Côte d'Ivoire (from 5.59% to 5.90%, RR 1.05, 304 deaths per 100,000, 118 excess deaths), Mozambique (from 5.78% to 6.11%, RR 1.06, 328 deaths per 100,000, 928 excess deaths), and Uganda (from 5.39% to 5.60%, RR 1.04, 209 deaths per 100,000, 466 excess deaths); Table 1 and Fig 2. Similar effects were observed for 6-, 9-, and 12-month strategies: In each case, predicted infant mortality increased compared to the current policy, with the number of excess deaths lower with longer durations of cotrimoxazole prophylaxis (Table 1 and Fig 2).

A programmatic strategy of no routine prophylaxis, and instead initiating cotrimoxazole only once HIV is diagnosed, led to the greatest increase in predicted mortality across settings: Zimbabwe (increase in predicted mortality from 5.58% to 5.84%, RR 1.05, predicted excess mortality rate 260 deaths per 100,000 infants HIV-exposed, 352 excess annual deaths), Côte d'Ivoire (from 5.59% to 5.91%, RR 1.06, 322 deaths per 100,000, or 125 excess deaths), Mozambique (from 5.78% to 6.12%, RR 1.06, 339 deaths per 100,000, or 961 excess deaths), and Uganda (from 5.39% to 5.61%, RR 1.04, 221 deaths per 100,000, 491 excess deaths); Table 1.

### Sensitivity analyses

Sensitivity analyses for Mozambique are shown in Fig 3. Reducing cotrimoxazole uptake to 40%, predicted excess mortality was lower overall, but the fewest predicted deaths still occurred under the current strategy of universal cotrimoxazole provision. Lowering the efficacy of cotrimoxazole in 5% intervals, to as little as 15% predicted mortality reduction (compared to 43% in the CHAP trial), the current strategy continued to have the lowest number of

**Table 1. Comparison of predicted mortality, between ages 6 weeks to 2 years, with different programmatic cotrimoxazole strategies for children born to mothers with HIV.** For the current WHO strategy (base case) of providing cotrimoxazole (CTX) to all HIV-exposed children, mortality rate is expressed in percent (%). Alternative strategies explored were (1) initiating cotrimoxazole at 6 weeks and stopping at 12 months; (2) initiating cotrimoxazole at 6 weeks and stopping at 9 months; (3) initiating cotrimoxazole at 6 weeks and stopping at 6 months; (4) initiating cotrimoxazole at 6 weeks and stopping at 3 months; and (5) initiating cotrimoxazole prophylaxis only once a positive HIV test result is confirmed. Mortality for alternative strategies are reported in comparison to the base case (current strategy): risk ratio, excess mortality rate per 100,000 HIV-exposed infants, and excess annual deaths. CTX = cotrimoxazole.

| | | Current | Programmatic strategy | | | | |
|---|---|---|---|---|---|---|---|
| | | | Alternate | | | | |
| | | | CTX from 6 weeks to 12 months | CTX from 6 weeks to 9 months | CTX from 6 weeks to 6 months | CTX from 6 weeks to 3 months | Start CTX only after positive HIV test result |
| Mozambique | Mortality rate (%) | 5.78 | 5.91 | 5.95 | 6.07 | 6.11 | 6.12 |
| | Risk ratio | - | 1.02 | 1.03 | 1.05 | 1.06 | 1.06 |
| | Excess mortality rate (per 100,000) | - | 131 | 174 | 295 | 328 | 339 |
| | Excess deaths | - | 371 | 492 | 835 | 928 | 961 |
| Cote d'Ivoire | Mortality rate (%) | 5.59 | 5.72 | 5.76 | 5.85 | 5.90 | 5.91 |
| | Risk ratio | - | 1.02 | 1.03 | 1.05 | 1.06 | 1.06 |
| | Excess mortality rate (per 100,000) | - | 127 | 172 | 254 | 304 | 322 |
| | Excess deaths | - | 49 | 67 | 98 | 118 | 125 |
| Zimbabwe | Mortality rate (%) | 5.58 | 5.68 | 5.71 | 5.79 | 5.83 | 5.84 |
| | Risk ratio | - | 1.02 | 1.02 | 1.04 | 1.04 | 1.05 |
| | Excess mortality rate (per 100,000) | - | 99 | 134 | 208 | 246 | 260 |
| | Excess deaths | - | 135 | 181 | 281 | 334 | 352 |
| Uganda | Mortality rate (%) | 5.39 | 5.48 | 5.51 | 5.57 | 5.60 | 5.61 |
| | Risk ratio | - | 1.02 | 1.02 | 1.03 | 1.04 | 1.04 |
| | Excess mortality rate (per 100,000) | - | 87 | 117 | 177 | 209 | 221 |
| | Excess deaths | - | 193 | 261 | 394 | 466 | 491 |

predicted deaths. When EID coverage was increased to 100%, the current strategy still had the lowest predicted mortality, compared to alternative approaches of shorter durations of cotrimoxazole, although predicted excess mortality reduced overall. To explore the effect of reaching and exceeding MTCT elimination targets, we reduced rates for both perinatal and postnatal vertical transmission to zero. Finally, a combined conservative estimate of low cotrimoxazole coverage (40%) and efficacy (15%), low vertical transmission (2%) and high EID (100%), designed to model a plausible future scenario in each country, showed the current strategy still led to fewer predicted deaths, though effects were attenuated.

Predicted mortality effects remained consistent across countries in all our sensitivity analyses. Mozambique has the highest vertical transmission rate of the 4 countries (13.5%) and high EID coverage (82.9%). If uptake was as low as 40% in Mozambique, starting cotrimoxazole only after a positive HIV test is estimated to result in 369 excess annual deaths compared to the current strategy (Fig 3). Figures for sensitivity analysis in Cote d'Ivoire (Fig G in S1 Text), Zimbabwe (Fig E in S1 Text), and Uganda (Fig J in S1 Text) are available in S1 Text. Uganda has the lowest vertical transmission rate (5.9%) and EID coverage of 66.2%. Further reducing perinatal and postnatal MTCT without changing EID is estimated to result in 311 and 251 excess annual deaths, respectively, if a strategy of starting cotrimoxazole only after a positive HIV test was deployed. Cote d'Ivoire has the lowest EID coverage of the 4 countries (60.8%) and vertical transmission of 7.8%. In the most optimistic consideration, if EID coverage was

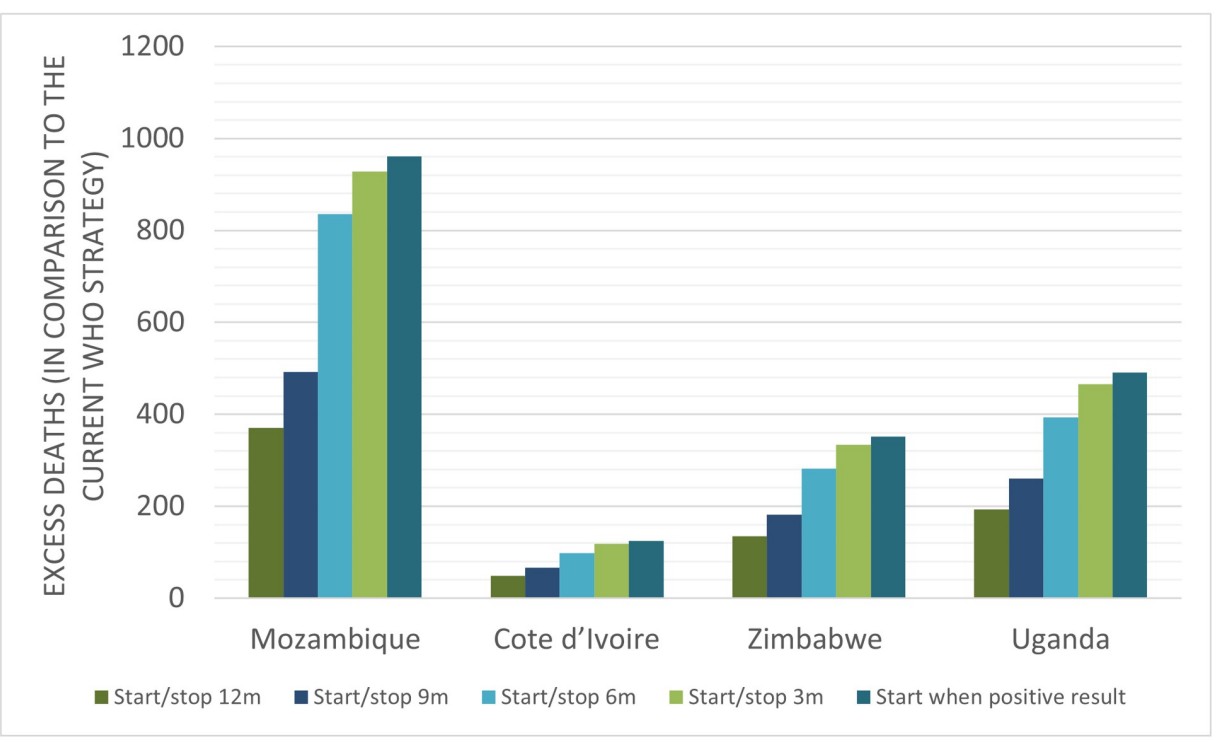

**Fig 2. Excess deaths per year by country under alternative cotrimoxazole strategies.** Columns represent additional deaths from each alternate strategy in comparison to the current WHO programmatic strategy of providing cotrimoxazole to all HIV-exposed infants. 12m = 12 months, 9m = 9 months, 6m = 6 months, 3m = 3 months.

increased to 100%, 3-month provision of cotrimoxazole is predicted to result in 51 excess annual deaths, in part due to continued vertical transmission through breastfeeding. In Zimbabwe, where the vertical transmission rate (8.7%) is similar to the regional estimate for Eastern and Southern Africa, a conservative ascribed mortality reduction to cotrimoxazole of 15% would still result in 352 excess annual deaths when cotrimoxazole is only started after a positive HIV test result.

## Discussion

Our study suggests that changing the current cotrimoxazole policy for children born to mothers with HIV is predicted to result in more deaths across the 4 modelled high-burden settings representing different epidemic contexts. The highest increase in mortality is predicted to occur with no routine cotrimoxazole provision (ranging from predicted excess mortality rate of 221 per 100,000 for Uganda to 339 per 100,000 for Mozambique). These predicted effects persisted, but were attenuated, when modelling lower cotrimoxazole uptake, smaller mortality benefits from prophylaxis, higher EID coverage, and lower perinatal and postnatal vertical transmission. These effects are driven entirely by the benefits of cotrimoxazole for infants who acquire HIV. Even as the epidemic context matures and most infants now remain HIV-free, many countries still have slow progress towards elimination targets. This study demonstrates the important "safety net" that cotrimoxazole may provide for children who are missed by the test-to-treat cascade in these settings.

Recent studies have questioned the ongoing value of cotrimoxazole in settings where most children born to women with HIV now remain HIV-free [8–10]. Our study shows that across

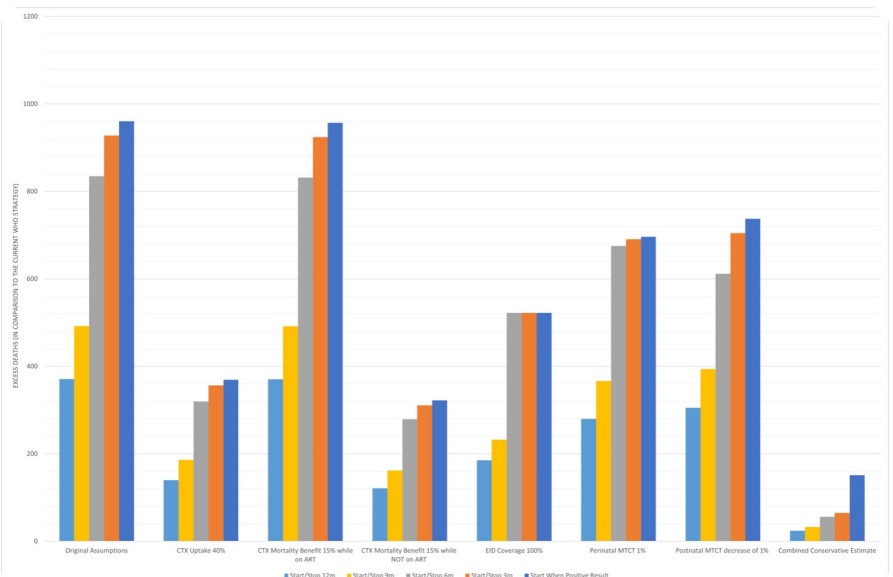

**Fig 3. Sensitivity analysis (Mozambique)—Conservative scenarios.** Sensitivity analysis exploring the effect of varying assumptions on the RR for deaths (6 weeks to 2 years) compared to the current WHO strategy for Mozambique: (1) Cotrimoxazole uptake at 40% rather than 100% assumed in the base model, (2) mortality reduction from cotrimoxazole while on ART decreased from 43% to 15%, (3) mortality reduction from cotrimoxazole while not on ART decreased from 43% to 15%, (4) EID coverage increased to 100% rather than 82.9% EID coverage for Mozambique, (5) perinatal MTCT reduced from to 6% to 1%, (6) postnatal MTCT decreased from 7.5% to 6.5%. (7) Conservative estimate concurrently combining sensitivity scenarios 1–6. CTX, cotrimoxazole; ART, antiretroviral therapy; EID, early infant diagnosis at 6 weeks; MTCT, mother-to-child transmission; 12m, 12 months; 9m, 9 months; 6m, 6 months; 3m, 3 months; RR, risk ratio; WHO, World Health Organization.

these 4 modelled settings, predicted mortality would increase by limiting provision of cotrimoxazole. When comparing modelled strategies, the highest predicted mortality occurred under an alternative strategy of providing no universal prophylaxis and instead starting cotrimoxazole only after a positive infant HIV test result. This is driven by high predicted mortality among infants with HIV who are not picked up by testing, and therefore not started on ART or cotrimoxazole [6,7]. Predicted mortality increased to different degrees when cotrimoxazole was given universally but for a shorter duration than currently. This is due to inadequate EID coverage, vertical transmission through breastfeeding after an initial negative test, and shortening cotrimoxazole duration during a period when disease progression among children with HIV remains rapid [3]. While the benefits of retaining the current strategy persisted even with suboptimal cotrimoxazole coverage, reflecting the reality in many settings, there were fewer benefits with low coverage, emphasising the continued effort needed to retain infants in care [20]. When comparing countries, Mozambique had the highest predicted excess mortality and greatest number of excess annual deaths under alternative cotrimoxazole strategies. This likely reflects high perinatal and postnatal transmission in Mozambique compared to the other modelled countries. Conversely, Uganda has the lowest MTCT rates, having made significant progress towards the elimination of paediatric HIV, and lowest predicted increase in mortality with alternative cotrimoxazole strategies. Mozambique has the highest EID testing rate but also had the greatest estimated increase in predicted mortality in our models [16]. Taken together, this suggests that while increased testing coverage is advantageous, continued focus should be placed on interventions to reduce vertical transmission. Where vertical transmission does occur, timely testing and initiation of antiretroviral treatment remains a cornerstone of care.

Countries with ongoing high rates of transmission would be most affected by any change in cotrimoxazole strategy.

There are several strengths and limitations to the analysis. Our model was developed with up-to-date, publicly available epidemiological estimates. These were supplemented by data extracted from the highest available levels of evidence. The model incorporates multiple aspects of transmission, testing and treatment uptake and mortality, and has been developed in an accessible format which can be adapted to other settings. Assumptions in the model were interrogated in sensitivity analyses. Although our effects of cotrimoxazole were derived from the CHAP trial—which enrolled children after infancy in the pre-ART era—fewer predicted deaths occurred under the current strategy even when the estimated mortality reduction from cotrimoxazole was as low as 15%. There are limited randomised controlled trial data for cotrimoxazole among breastfed infants with HIV, but it is unlikely that mortality benefits are substantially lower than in the original CHAP trial, given the more rapid disease progression at this age, lack of prognostic markers, and higher risk *of Pneumocystis jirovecii* pneumonia [9]. Our sensitivity analysis shows that, even if this assumption is untrue and the benefits are much lower, the current policy leads to fewer predicted deaths than a revised policy of shorter (or no) prophylaxis. There are also limitations to this analysis. There are several parameters for which data were not available (e.g., the probability of having a 9-month test if early infant diagnosis has been missed). We assumed equal uptake of cotrimoxazole regardless of final child HIV status, but it is likely that children most likely to acquire HIV are the least engaged in care and are therefore less likely to receive cotrimoxazole. Countries achieving 95-95-95 testing and treatment targets and Path to Elimination of MTCT goals were not modelled and may no longer require routine cotrimoxazole given that few infants acquire HIV and those who do are most likely to be missed by testing [21].

We only modelled mortality increases following reductions in cotrimoxazole provision, given the known benefits of cotrimoxazole for children with HIV; however, our goal was to quantify the magnitude of mortality increase for each strategy across epidemic contexts to provide policymakers with population estimates when considering guideline revisions. We recognise, however, that our model did not incorporate the potential risks of cotrimoxazole, such as antimicrobial resistance, as mortality effects are difficult to quantify given limited quality data from sub-Saharan Africa [22]. There is evidence that cotrimoxazole use in infants who are HEU leads to antimicrobial resistance, including 1 small study showing cross-class resistance to amoxicillin, but no evidence of major microbiome dysbiosis from available data [23]. Although there are estimates of deaths attributable to resistance for specific antibiotics in several global regions [24], it is difficult to model what proportion of these deaths would be averted in each of our scenarios, particularly as it remains unclear whether cotrimoxazole resistance is reversible to a clinically relevant level with decline in use [25]. Estimates of pathogen-specific deaths attributable to AMR from cotrimoxazole in sub-Saharan Africa in the post-neonatal period range from 0.0 (0.0 to 0.1) per 100,000 for *Proteus* spp. to 15.07 (2.3 to 31.8) per 100,000 for *Streptococcus pneumoniae*, but even collectively these estimates are much lower than the excess deaths modelled in our programmatic scenarios from restricted cotrimoxazole use. Responsive country-level surveillance is required to evaluate local resistance data in the context of evolving HIV epidemics. Lastly, this study did not consider the economic implications of the proposed strategies. Future research should investigate the potential cost savings and the financial impacts of implementing each strategy on national budgets [26].

Countries need to continue to strengthen the test-to-treat cascade, to ensure timely testing and ART initiation, in order to reduce morbidity and mortality among infants who are HIV-exposed. However, our models show that cotrimoxazole still remains an important part of the package of care. Decisions around appropriate provision of cotrimoxazole to infants who are

HIV-exposed require a pragmatic, public health approach with difficult decisions that weigh risks and benefits that differ across settings. Results from this model may be considered as part of the breadth of evidence which help inform policymaker decisions. Though there has been progress in prevention of paediatric HIV, the "last mile" to elimination is particularly challenging. Furthermore, in many high-burden settings, testing coverage is not universal, making it difficult to reliably distinguish infants who are HIV-infected or HEU. Individualised algorithms dependent on testing contingencies may be unrealistic given resource and service limitations at most low-level health facilities. The current programmatic cotrimoxazole strategy provides important benefits for the small number of high-risk infants who still acquire HIV through vertical transmission and are missed due to gaps in the test-to-treat cascade. As demonstrated in the sensitivity analysis for each strategy, lower vertical transmission rates and improved testing and treatment uptake attenuate the predicted increase in mortality. This demonstrates the importance of considering how vertical transmission should inform cotrimoxazole policy: in the context of high ongoing transmission, the current universal cotrimoxazole prophylaxis approach is predicted to lead to substantially fewer deaths than alternative strategies, while settings with low transmission rates may derive less benefit from universal cotrimoxazole prophylaxis. Therefore, hybrid cotrimoxazole policies may be a consideration noting we did not model countries achieving HIV elimination targets or with strong test-to-treat cascades. Further research is required to better understand cotrimoxazole coverage for children who are HIV-exposed; to determine the proportion who are subsequently found to be living with HIV; to better understand community values and preferences; and to identify barriers to engagement with the test-to-treat cascade. Finally, cost-effectiveness analysis informed by national programme cost data is required to better understand the most optimal health investment and cost per death averted and economic impact of each strategy.

In conclusion, despite impressive progress in prevention of vertical HIV transmission, elimination is not yet in sight in most high-burden settings. Priorities include maintaining optimal care and ART for mothers; providing postnatal HIV prophylaxis for infants; and ensuring timely testing and linkage to care. However, cotrimoxazole continues to provide a vital safety net for infants who acquire HIV but are not tested or started on ART. Our data suggest that changing the current strategy of universal cotrimoxazole provision for all infants who are HIV-exposed would increase predicted mortality to varying degrees across the 4 modelled countries, depending on vertical transmission rates and test-to-treatment cascade coverage. Policymakers need to balance the possible negative effects of cotrimoxazole with potential excess deaths resulting from changes in programmatic use depending on their specific context.

## Supporting information

**S1 TRIPOD Checklist. TRIPOD Checklist: Prediction model development.**
(DOCX)

**S1 Text.  Table A in S1 Text. Model assumptions and data sources.** CTX = cotrimoxazole, ART = antiretroviral therapy, EID = early infant diagnosis at 6 weeks, ZIM = Zimbabwe, CIV = Cote d'Ivoire, MOZ = Mozambique, UGA = Uganda. *The authorship brings multidisciplinary expertise in paediatric HIV (AJP, CE, DMG, CW, MP), epidemiology (MS, DMG), clinical trials (DMG, AJP, MP, MS), and policy (SM, MP). **Fig A in S1 Text. Predicted mortality percentage per year by country under alternative cotrimoxazole strategies.** Columns represent additional deaths from each alternate strategy in comparison to the current WHO programmatic strategy of providing cotrimoxazole to all HIV-exposed infants. 12m = 12 months, 9m = 9 months, 6m = 6 months, 3m = 3 months. **Fig B in S1 Text. Predicted excess mortality rate (per 100,000) per year by country under alternative cotrimoxazole**

**strategies.** Columns represent additional deaths from each alternate strategy in comparison to the current WHO programmatic strategy of providing cotrimoxazole to all HIV-exposed infants. 12m = 12 months, 9m = 9 months, 6m = 6 months, 3m = 3 months. **Fig C in S1 Text. Predicted risk ratio per year by country under alternative cotrimoxazole strategies.** Columns represent additional deaths from each alternate strategy in comparison to the current WHO programmatic strategy of providing cotrimoxazole to all HIV-exposed infants. 12m = 12 months, 9m = 9 months, 6m = 6 months, 3m = 3 months. **Fig D in S1 Text. Sensitivity Analysis for Zimbabwe (risk ratios).** Sensitivity analysis, for Zimbabwe, exploring the effect of varying assumptions on the risk ratio for deaths (6 weeks to 2 years) compared to the current WHO strategy. CTX = cotrimoxazole, ART = antiretroviral therapy, EID = early infant diagnosis at 6 weeks, MTCT = mother-to-child transmission, 12m = 12 months, 9m = 9 months, 6m = 6 months, 3m = 3 months. **Fig E in S1 Text. Sensitivity Analysis for Zimbabwe (excess deaths).** Sensitivity analysis, for Zimbabwe, exploring the effect of varying assumptions on the risk ratio for deaths (6 weeks to 2 years) compared to the current WHO strategy. CTX = cotrimoxazole, ART = antiretroviral therapy, EID = early infant diagnosis at 6 weeks, MTCT = mother-to-child transmission, 12m = 12 months, 9m = 9 months, 6m = 6 months, 3m = 3 months. **Fig F in S1 Text. Sensitivity Analysis for Cote d'Ivoire (risk ratios).** Sensitivity analysis, for Cote d'Ivoire, exploring the effect of varying assumptions on the risk ratio for deaths (6 weeks to 2 years) compared to the current WHO strategy. CTX = cotrimoxazole, ART = antiretroviral therapy, EID = early infant diagnosis at 6 weeks, MTCT = mother-to-child transmission, 12m = 12 months, 9m = 9 months, 6m = 6 months, 3m = 3 months. **Fig G in S1 Text. Sensitivity Analysis for Cote d'Ivoire (excess deaths).** Sensitivity analysis, for Cote d'Ivoire, exploring the effect of varying assumptions on the risk ratio for deaths (6 weeks to 2 years) compared to the current WHO strategy. CTX = cotrimoxazole, ART = antiretroviral therapy, EID = early infant diagnosis at 6 weeks, MTCT = mother-to-child transmission, 12m = 12 months, 9m = 9 months, 6m = 6 months, 3m = 3 months. **Fig H in S1 Text. Sensitivity Analysis for Mozambique (risk ratios).** Sensitivity analysis, for Mozambique, exploring the effect of varying assumptions on the risk ratio for deaths (6 weeks to 2 years) compared to the current WHO strategy. CTX = cotrimoxazole, ART = antiretroviral therapy, EID = early infant diagnosis at 6 weeks, MTCT = mother-to-child transmission, 12m = 12 months, 9m = 9 months, 6m = 6 months, 3m = 3 months. **Fig I in S1 Text. Sensitivity Analysis for Uganda (risk ratios).** Sensitivity analysis, for Uganda, exploring the effect of varying assumptions on the risk ratio for deaths (6 weeks to 2 years) compared to the current WHO strategy. CTX = cotrimoxazole, ART = antiretroviral therapy, EID = early infant diagnosis at 6 weeks, MTCT = mother-to-child transmission, 12m = 12 months, 9m = 9 months, 6m = 6 months, 3m = 3 months. **Fig J in S1 Text. Sensitivity Analysis for Uganda (excess deaths).** Sensitivity analysis, for Uganda, exploring the effect of varying assumptions on the risk ratio for deaths (6 weeks to 2 years) compared to the current WHO strategy. CTX = cotrimoxazole, ART = antiretroviral therapy, EID = early infant diagnosis at 6 weeks, MTCT = mother-to-child transmission, 12m = 12 months, 9m = 9 months, 6m = 6 months, 3m = 3 months. **Fig K in S1 Text. Sensitivity Variable—Cotrimoxazole uptake, Zimbabwe.** Risk ratio of mortality for varying risk reduction from cotrimoxazole uptake for Zimbabwe, (risk ratio). CTX = cotrimoxazole, 12m = 12 months, 9m = 9 months, 6m = 6 months, 3m = 3 months. **Fig L in S1 Text. Sensitivity Variable–Risk reduction from CTX, Zimbabwe.** Risk ratio of mortality for varying risk reduction from cotrimoxazole while infant with HIV is taking antiretroviral therapy for Zimbabwe, (risk ratio). Risk reduction from 15% to 43%. CTX = cotrimoxazole, ART = antiretroviral therapy, 12m = 12 months, 9m = 9 months, 6m = 6 months, 3m = 3 months. **Fig M in S1 Text. Sensitivity Variable—Risk reduction from CTX, Zimbabwe.** Risk ratio of mortality for varying risk reduction from

cotrimoxazole while infant with HIV is taking antiretroviral therapy for Zimbabwe, (risk ratio). Risk reduction from 25% to 60%. CTX = cotrimoxazole, ART = antiretroviral therapy, 12m = 12 months, 9m = 9 months, 6m = 6 months, 3m = 3 months. **Fig N in S1 Text. Sensitivity Variable—EID testing, Zimbabwe.** Risk ratio of mortality for varying probability of HIV-exposed infants undergoing early infant diagnosis (EID) for Zimbabwe, (risk ratio). EID = early infant diagnosis at 6 weeks, 12m = 12 months, 9m = 9 months, 6m = 6 months, 3m = 3 months. **Fig O in S1 Text. Sensitivity Variable—Perinatal MTCT, Zimbabwe.** Risk ratio of mortality for varying probability of perinatal mother-to-child transmission (MTCT) for Zimbabwe, (risk ratio). MTCT = mother-to-child transmission, 12m = 12 months, 9m = 9 months, 6m = 6 months, 3m = 3 months. **Fig P in S1 Text. Sensitivity Variable—Postnatal MTCT, Zimbabwe.** Risk ratio of mortality for varying probability of postnatal mother-to-child transmission (MTCT) for Zimbabwe, (risk ratio). MTCT = mother-to-child transmission, 12m = 12 months, 9m = 9 months, 6m = 6 months, 3m = 3 months. **Fig Q in S1 Text. Sensitivity Variable—Cotrimoxazole uptake, Cote d'Ivoire.** Risk ratio of mortality for varying risk reduction from cotrimoxazole uptake for Cote d'Ivoire, (risk ratio). CTX = cotrimoxazole, 12m = 12 months, 9m = 9 months, 6m = 6 months, 3m = 3 months. **Fig R in S1 Text. Sensitivity Variable—Risk reduction from CTX, Cote d'Ivoire.** Risk ratio of mortality for varying risk reduction from cotrimoxazole while infant with HIV is taking antiretroviral therapy for Cote d'Ivoire, (risk ratio). Risk reduction from 15% to 43%. CTX = cotrimoxazole, ART = antiretroviral therapy, 12m = 12 months, 9m = 9 months, 6m = 6 months, 3m = 3 months. **Fig S in S1 Text. Sensitivity Variable—Risk reduction from CTX, Cote d'Ivoire.** Risk ratio of mortality for varying risk reduction from cotrimoxazole while infant with HIV is taking antiretroviral therapy for Cote d'Ivoire, (risk ratio). Risk reduction from 15% to 60%. CTX = cotrimoxazole, ART = antiretroviral therapy, 12m = 12 months, 9m = 9 months, 6m = 6 months, 3m = 3 months. **Fig T in S1 Text. Sensitivity Variable—EID testing, Cote d'Ivoire.** Risk ratio of mortality for varying probability of HIV-exposed infants undergoing early infant diagnosis (EID) for Cote d'Ivoire, (risk ratio). EID = early infant diagnosis at 6 weeks, 12m = 12 months, 9m = 9 months, 6m = 6 months, 3m = 3 months. **Fig U in S1 Text. Sensitivity Variable—Perinatal MTCT, Cote d'Ivoire.** Risk ratio of mortality for varying probability of perinatal mother-to-child transmission (MTCT) for Cote d'Ivoire, (risk ratio). MTCT = mother-to-child transmission, 12m = 12 months, 9m = 9 months, 6m = 6 months, 3m = 3 months. **Fig V in S1 Text. Sensitivity Variable—Postnatal MTCT, Cote d'Ivoire.** Risk ratio of mortality for varying probability of postnatal mother-to-child transmission (MTCT) for Cote d'Ivoire, (risk ratio). MTCT = mother-to-child transmission, 12m = 12 months, 9m = 9 months, 6m = 6 months, 3m = 3 months. **Fig W in S1 Text. Sensitivity Variable—Cotrimoxazole uptake, Mozambique.** Risk ratio of mortality for varying risk reduction from cotrimoxazole uptake for Mozambique, (risk ratio). CTX = cotrimoxazole, 12m = 12 months, 9m = 9 months, 6m = 6 months, 3m = 3 months. **Fig X in S1 Text. Sensitivity Variable—Risk reduction from CTX, Mozambique.** Risk ratio of mortality for varying risk reduction from cotrimoxazole while infant with HIV is taking antiretroviral therapy for Mozambique, (risk ratio). Risk reduction from 15% to 43%. CTX = cotrimoxazole, ART = antiretroviral therapy, 12m = 12 months, 9m = 9 months, 6m = 6 months, 3m = 3 months. **Fig Y in S1 Text. Sensitivity Variable—Risk reduction from CTX, Mozambique.** Risk ratio of mortality for varying risk reduction from cotrimoxazole while infant with HIV is taking antiretroviral therapy for Mozambique, (risk ratio). Risk reduction from 15% to 60%. CTX = cotrimoxazole, ART = antiretroviral therapy, 12m = 12 months, 9m = 9 months, 6m = 6 months, 3m = 3 months. **Fig Z in S1 Text. Sensitivity Variable—EID testing, Mozambique.** Risk ratio of mortality for varying probability of HIV-exposed infants undergoing early infant diagnosis (EID) for Mozambique, (risk ratio). EID = early infant diagnosis at 6

weeks, 12m = 12 months, 9m = 9 months, 6m = 6 months, 3m = 3 months. **Fig AA in S1 Text. Sensitivity Variable—Perinatal MTCT, Mozambique.** Risk ratio of mortality for varying probability of perinatal mother-to-child transmission (MTCT) for Mozambique, (risk ratio). MTCT = mother-to-child transmission, 12m = 12 months, 9m = 9 months, 6m = 6 months, 3m = 3 months. **Fig AB in S1 Text. Sensitivity Variable—Postnatal MTCT, Mozambique.** Risk ratio of mortality for varying probability of postnatal mother-to-child transmission (MTCT) for Mozambique, (risk ratio). MTCT = mother-to-child transmission, 12m = 12 months, 9m = 9 months, 6m = 6 months, 3m = 3 months. **Fig AC in S1 Text. Sensitivity Variable—Cotrimoxazole uptake, Uganda.** Risk ratio of mortality for varying risk reduction from cotrimoxazole uptake for Uganda, (risk ratio). CTX = cotrimoxazole, 12m = 12 months, 9m = 9 months, 6m = 6 months, 3m = 3 months. **Fig AD in S1 Text. Sensitivity Variable—Risk reduction from CTX, Uganda.** Risk ratio of mortality for varying risk reduction from cotrimoxazole while infant with HIV is taking antiretroviral therapy for Uganda, (risk ratio). Risk reduction from 15% to 43%. CTX = cotrimoxazole, ART = antiretroviral therapy, 12m = 12 months, 9m = 9 months, 6m = 6 months, 3m = 3 months. **Fig AE in S1 Text. Sensitivity Variable—Risk reduction from CTX, Uganda.** Risk ratio of mortality for varying risk reduction from cotrimoxazole while infant with HIV is taking antiretroviral therapy for Uganda, (risk ratio). Risk reduction from 15% to 60%. CTX = cotrimoxazole, ART = antiretroviral therapy, 12m = 12 months, 9m = 9 months, 6m = 6 months, 3m = 3 months. **Fig AF in S1 Text. Sensitivity Variable—EID testing, Uganda.** Risk ratio of mortality for varying probability of HIV-exposed infants undergoing early infant diagnosis (EID) for Uganda, (risk ratio). EID = early infant diagnosis at 6 weeks, 12m = 12 months, 9m = 9 months, 6m = 6 months, 3m = 3 months. **Fig AG in S1 Text. Sensitivity Variable—Perinatal MTCT, Uganda.** Risk ratio of mortality for varying probability of perinatal mother-to-child transmission (MTCT) for Uganda, (risk ratio). MTCT = mother-to-child transmission, 12m = 12 months, 9m = 9 months, 6m = 6 months, 3m = 3 months. **Fig AH in S1 Text. Sensitivity Variable—Postnatal MTCT, Uganda.** Risk ratio of mortality for varying probability of postnatal mother-to-child transmission (MTCT) for Uganda, (risk ratio). MTCT = mother-to-child transmission, 12m = 12 months, 9m = 9 months, 6m = 6 months, 3m = 3 months. (DOCX)

## Acknowledgments

Disclaimer: The content of this publication is solely the responsibility of the authors and does not necessarily represent the official views of the World Health Organization.

## Author Contributions

**Conceptualization:** Ceri Evans, Catherine J. Wedderburn, Diana M. Gibb, Martina Penazzato, Andrew J. Prendergast.

**Data curation:** Shrey Mathur.

**Formal analysis:** Shrey Mathur, Melanie Smuk, Andrew J. Prendergast.

**Methodology:** Shrey Mathur, Melanie Smuk, Ceri Evans, Catherine J. Wedderburn, Diana M. Gibb, Martina Penazzato, Andrew J. Prendergast.

**Supervision:** Andrew J. Prendergast.

**Validation:** Shrey Mathur, Melanie Smuk, Andrew J. Prendergast.

**Visualization:** Melanie Smuk.

**Writing – original draft:** Shrey Mathur, Melanie Smuk, Andrew J. Prendergast.

**Writing – review & editing:** Shrey Mathur, Melanie Smuk, Ceri Evans, Catherine J. Wedderburn, Diana M. Gibb, Martina Penazzato, Andrew J. Prendergast.

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
