## [Editor Report · Decision Letter 0]

13 Jun 2023

Dear Dr Mathur, 

Thank you for submitting your manuscript entitled "The Impact of Alternative Programmatic Cotrimoxazole Strategies on Mortality Among Children Born to Mothers with HIV: A Modelling Study" for consideration by PLOS Medicine.

Your manuscript has now been evaluated by the PLOS Medicine editorial staff and I am writing to let you know that we would like to send your submission out for external peer review.

Please re-submit your manuscript within two working days, i.e. by Jun 15 2023 11:59PM.

Kind regards,

Alexandra Schaefer, PhD

Associate Editor

PLOS Medicine

---

## [Decision Letter · Decision Letter 1]

23 Aug 2023

Dear Dr. Mathur,

Thank you very much for submitting your manuscript "The Impact of Alternative Programmatic Cotrimoxazole Strategies on Mortality Among Children Born to Mothers with HIV: A Modelling Study" (PMEDICINE-D-23-01623R1) for consideration at PLOS Medicine. 

Your paper was evaluated by four independent reviewers, including a statistical reviewer, and discussed among all the editors here and with an academic editor with relevant expertise. The reviews are appended at the bottom of this email and any accompanying reviewer attachments can be seen via the link below:

[LINK]

In light of these reviews, I am afraid that we will not be able to accept the manuscript for publication in the journal in its current form, but we would like to consider a revised version that addresses the reviewers' and editors' comments. Obviously we cannot make any decision about publication until we have seen the revised manuscript and your response, and we plan to seek re-review by one or more of the reviewers. 

We expect to receive your revised manuscript by Sep 13 2023 11:59PM. Please email us (plosmedicine@plos.org) if you have any questions or concerns.

We look forward to receiving your revised manuscript. 

Sincerely,

Louise Gaynor-Brook, MBBS PhD

plosmedicine.org

General comments:

Throughout the paper, please adapt reference call-outs to the following style: "... HIV-free [4,5]." (noting the absence of spaces within the square brackets).

Please revise ‘impact’ to ‘potential impact’ or similar when referring to your own results

Throughout the main text, please refer to ‘predicted mortality’ when referring to your own results

Data availability:

PLOS Medicine requires that the de-identified data underlying the specific results in a published article be made available, without restrictions on access, in a public repository or as Supporting Information at the time of article publication, provided it is legal and ethical to do so. If the data are freely or publicly available, please note this and state the location of the data (include the DOI or accession number); please note that a study author cannot be responsible for permitting access to the data or be the contact person for the data (I note that Andrew Prendergast approved my access request on OSF). 

Title: Please revise your title according to PLOS Medicine's style. We suggest “Estimating the Impact of Alternative Programmatic Cotrimoxazole Strategies on Mortality Among Children Born to Mothers with HIV: A Modelling Study” or similar

Abstract:

Line 48 - please revise to ‘potential impact’

In the last sentence of the Abstract Methods and Findings section, please describe 2-3 of the main limitations of the study's methodology.

Abstract Conclusions: Please address the implications of your study, being careful to avoid assertions of primacy. 

Author Summary:

In the final bullet point of ‘What Do These Findings Mean?’, please describe the main limitations of the study in non-technical language.

Introduction:

If there has been a systematic review of the evidence related to your study (or you have conducted one), please refer to and reference that review and indicate whether it supports the need for your study. 

Methods:

Did your study have a prospective protocol or analysis plan? If so, please include the relevant prospectively written document with your revised manuscript as a Supporting Information file to be published alongside your study, and cite it in the Methods section. Changes in the analysis-- including those made in response to peer review comments-- should be identified as such in the Methods section of the paper, with rationale. 

We suggest that TRIPOD may be an appropriate checklist for your study design; please include a completed checklist as a supplementary file. Please add the following statement, or similar, to the Methods: "This study is reported as per the Transparent reporting of a multivariable prediction model for individual prognosis or diagnosis (TRIPOD) guideline (S1 Checklist)." The TRIPOD guideline can be found here: https://www.equator-network.org/reporting-guidelines/tripod-statement/ When completing the checklist, please use section and paragraph numbers, rather than page numbers which will likely no longer correspond to the appropriate sections after copy-editing.

Using guidance from Geoffrey P Garnett, Simon Cousens, Timothy B Hallett, Richard Steketee, Neff Walker. Mathematical models in the evaluation of health programmes. (2011) Lancet DOI:10.1016/S0140-6736(10)61505-X; 

Please provide a diagram that shows the model structure, including how the disease natural history is represented and how the putative intervention (cotrimoxazole) could affect the system.

Please provide a complete list of model parameters and important caveats about the use of these values noted.

Please provide a clear statement about how the model was fitted to the data (may include goodness-of-fit measure, the numerical algorithm used, which parameter varied, constraints imposed on parameter values, and starting conditions)

For uncertainty analyses, please state the sources of uncertainties quantified and not quantified [can include parameter, data, and model structure].

Please discuss the scientific rationale for this choice of model structure and identify points where this choice could influence conclusions drawn. Please also describe the strength of the scientific basis underlying the key model assumptions.

Results: 

Line 236 - please clarify that sensitivity analyses for Mozambique are shown in Fig 3

Throughout the results section, please refer to ‘predicted mortality’ 

Please provide tables/figures for all of the sensitivity analyses presented in the main text (these can be included in the supplementary information)

Line 252 - please refer to specific supplementary figures for each of these countries. NB) these supplementary figures have not been included in this submission.

Discussion:

Please present and organize the Discussion as follows: a short, clear summary of the article's findings; what the study adds to existing research and where and why the results may differ from previous research; strengths and limitations of the study; implications and next steps for research, clinical practice, and/or public policy; one-paragraph conclusion.

Throughout the discussion, please refer to ‘predicted mortality’ when referring to your own results

Figures:

Please consider avoiding the use of red and green in order to make your figure more accessible to those with colour blindness.

Please define all abbreviations used in the figure legend of each figure.

Tables:

Please define all abbreviations used in the table legend of each table (including supplementary tables).

References:

Please ensure that journal name abbreviations match those found in the National Center for Biotechnology Information (NCBI) databases (http://www.ncbi.nlm.nih.gov/nlmcatalog/journals), and are appropriately formatted and capitalised.

Please also see https://journals.plos.org/plosmedicine/s/submission-guidelines#loc-references for further details on reference formatting. 

Where website addresses are cited, please specify the date of access. 

Comments from the reviewers:

Reviewer #1: Mathur and colleagues describe in this manuscript deterministic compartmental models developed by the authors relying upon up-to-date, publicly available estimates including of HIV early infant HIV testing (EIT), prevalence of infant HIV acquisition, and antiretroviral treatment (ART) uptake to evaluate mortality between 6- and 24-months of life among HIV-exposed children comparing a strategy of universal cotrimoxazole use for all children to other strategies with lower coverage, as cotrimoxazole has not bene shown to have any survival benefit when used in children who are HIV-exposed uninfected, but has been shown to improve survival in children living with HIV not yet on treatment. Models were applied to four high burden HIV-countries yet to achieve 95-95-95 HIV program targets or Eliminating Mother-to-Child-Transmission goals, specifically Cote d'Ivoire, Mozambique, Uganda, and Zimbabwe. Country-specific data was supplemented with data extracted from peer-reviewed publications of clinical trials, observational studies, and other epidemiological data in eras both before and after universal ART was recommended at time of HIV diagnoses. The paper is well-written with excellent tables and figures. Considerable work has been devoted to testing multiple strategies. The authors conclude changing the current recommendations of universal cotrimoxazole provision for all HIV-exposed infants would increase mortality to varying degrees depending on vertical transmission rates and test-to-treatment cascade strategies would increase mortality to varying degrees. While this is an important manuscript that will be of interest to PLOS Medicine readers, there are two issues raised by the final recommendation of the modeling exercise that need to be addressed by the authors. Major and minor revision recommendations are presented below.

Major Revision Recommendations:

The authors' final conclusions advocate for universal cotrimoxazole use by HIV-exposed infants until infection is excluded and vertical transmission risk has ended. Conflictingly, the authors make mention in the discussion that among countries achieving 95-95-95 testing and treatment targets and Elimination of Mother-to-Child goals, cotrimoxazole prophylaxis may not be required. Interestingly, in at least one of the two studies cited for the lack of survival benefit of cotrimoxazole use among infants and children HIV-exposed uninfected, the Mpepu randomized controlled trial was conducted in a setting where the 95-95-95 targets have been achieved and infant HIV acquisition is currently estimated to be approximately 1%. This presents an argument for a hybrid policy, as the absence of using a country such as Botswana in the modeling exercise may be influencing an incorrect policy recommendation about universal use of cotrimoxazole.

The model assumes that 100% of infants who are HIV-exposed and for whom HIV infection status has not been finalized or vertical transmission risk continues to exist will be prescribed cotrimoxazole and parents or caregivers will have 100% adherence to dosing. While the authors acknowledged that they lacked a data source to estimate the number of children most likely to acquire HIV who are not engaged in care, this is a major flaw in the model that leads to questionable justification for continued universal cotrimoxazole use. If there is an issue with timely HIV testing of infants at risk for HIV, or timely ART initiation for those who found to be infected, monthly or quarterly encounters with the health care system for refills of cotrimoxazole are highly unlikely. If they are occurring, they could be better spent employing HIV testing or initiation of treatment, as applicable. It would be important in the Discussion section to call for accurate accounting of cotrimoxazole use in HIV-exposed infants to determine the proportion who actually receive this treatment and are subsequently found to be living with HIV, with a further understanding of the hindrances that preclude timely diagnoses and treatment initiation. The elephant in the room is engagement in the care cascade. To justify a policy that, on paper appears to save lives, but exchanges one medication regimen, namely ART, for another, cotrimoxazole, promulgates continued weaknesses in the treatment and care cascade and may only save lives on paper.

Minor Revision Recommendations:

Please use person first language throughout the manuscript. For example, at line 45 reference is made for "cotrimoxazole prophylaxis for HIV-exposed infants". Please revised this to read "cotrimoxazole prophylaxis for infants who are HIV-exposed".

Please review references, as some have "abbreviated" authorship. (eg references 1, 2, 15 and 20). 

At lines 349-350, reference is made to "EMTCT Path to Elimination goals". However, since the "E" in EMTCT stand for Elimination, this does not make sense. It might be clearer to take out the word "Elimination" in this phrase.

Reviewer #2: This paper presents the mortality effects of providing cotrimoxazole for periods shorter than the WHO guidelines. The conclusion is obvious, shorter periods of cotrim results in more deaths, but the paper is useful for showing the magnitude of the effect. 

1. In my opinion the paper is not really an evaluation of the WHO guidelines because it doe not consider any costs or negative effects. It would be relatively easy to show the additional costs of longer periods of cotrim and the cost per death averted. Since cotrim is inexpensive I assume it is very cost-effective. The authors do not include increased antimicrobial resistance because they are unsure of the effect size. But they did extensive sensitivity analysis of the model parameters. Why not include sensitivity around assumptions about resistance? It would make the paper much stronger.

2. The paper states that the model is freely available and provide a website. I tried to download the model but have been unsuccessful so far. I went to the OSF website using the link provided in the paper, but had to first create an account with OSF. Once I did that I received a message that I had to request permission to review the file. I requested permission. The next day I was informed that permission had been granted. When I went to the site I just saw a folder that had been set up for collaboration. There was no Excel file in that folder. So I still have not seen the model. Other readers may be frustrated with this process. Since I had to request permission there was no hope of a confidential review. 

Reviewer #3: The study appears to be highly relevant and valuable in informing the ongoing debate regarding the revision of the current WHO policy on administering cotrimoxazole to all HIV-exposed infants during breastfeeding. I believe that the research findings will serve as a credible and significant piece of evidence for understanding the disease and its treatment. Here are several suggested comments that the authors could address to enhance the manuscript:

1. The manuscript requires revisions to enhance clarity and readability. The authors should clarify certain sections to ensure their intended meaning is clear to the reader. Additionally, there are ambiguous statements that need to be revised for better comprehension. Coherence should be improved to ensure a logical flow of ideas between paragraphs. By addressing these specific areas for improvement, the authors can significantly enhance the overall quality of the manuscript, making it more accessible to readers. I have provided specific examples below:

a) Until the elimination of vertical transmission is achieved, therefore, children born to mothers with HIV require programmatic strategies to reduce morbidity and mortality.

b) There is therefore a critical need to identify and distinguish children with HIV from children who are HEU across different periods of transmission, given the distinct cotrimoxazole strategy required in each group.

c) Even where infants are tested, challenges remain across the testing-to-treatment cascade, with delay in the return of results, lack of integration with child health services, loss to follow-up, and delayed ART initiation.

d) The model starts at age 6 weeks when early infant diagnosis and cotrimoxazole initiation are recommended and estimates mortality to 24 months of age

e) Sensitivity analysis (results) paragraph one: "Reducing perinatal and postnatal vertical transmission rates towards elimination targets and beyond decreased the effect of cotrimoxazole on mortality until it reached zero effect at 0% transmission."

f) Once HIV was diagnosed (by 9 months or EoB), 76% mortality reduction from ART plus 43% mortality reduction from cotrimoxazole was applied, for the 77% of children with HIV who start ART

2. The authors referenced "expert opinion" as a data source in Supplementary Table 1. To improve clarity, it is recommended to provide additional details in the method section regarding the nature of the experts used and how "expert opinion" was utilised to extract data. Specifically, explain the qualifications or expertise of the experts involved and describe the process by which their opinions were obtained and incorporated into the analysis. Additionally, within the manuscript, it would be beneficial to clarify the assumptions ("Since data on subsequent testing are not available, we assumed that 80% of children undergoing EID would have a 9-month test, and 50% would have an EoB test, as these children would likely be engaged in the test-to-treatment cascade. For children not undergoing 162 EID at 6 weeks, we assumed the probability of subsequent testing was much lower (10% for 9-month 163 test, 30% for EoB), as these children were likely less engaged with services.") that were made for testing if they are based on expert opinion. This will help readers understand the methodology and the basis for the analysis, ensuring transparency and reproducibility.

3. The authors state that in all strategies, they assume that infants testing positive for HIV will initiate cotrimoxazole treatment. However, it would be beneficial for the manuscript to explicitly mention the probability of an HIV test yielding a positive result and provide clarification regarding the specific time points at which testing was assigned in the model. This additional information will enhance the transparency of the study and enable readers to better understand the assumptions made regarding HIV test outcomes and the timing of testing within the modeling.

4. If the following details mentioned inside the manuscript are linked with comment 3 please revise the manuscript accordingly. "To determine the probability of a child testing HIV-positive by age 6 weeks, perinatal transmission rates were obtained from country-specific UNAIDS estimates from the most recent year with available data for all countries, including six-week transmission rates (MOZ 6%, ZWE 152 5%, CIV 4%, and UGA 3%) (2). For postnatal transmission, we calculated the cumulative probability of acquiring HIV by age 9 months (MOZ 3.4%, ZWE 1.7%, CIV 1.7%, UGA 1.3%), and by the end of breastfeeding (MOZ 4.1%, ZWE 2.0%, CIV 2.1%, UGA 1.6%), based on derived weekly transmission rates from country-specific final vertical transmission rates and breastfeeding duration, making the assumption that postnatal transmission rates were consistent over time."

5. The authors mentioned in the manuscript that globally, 62% of infants currently receive the recommended virological test within two months of birth, and as a result, 40% of children with HIV remain undiagnosed. It would be helpful if the authors could clarify whether these specific values were utilized within the model being discussed in the study. Providing this clarification will enable readers to better understand the extent to which these statistics were incorporated into the modeling process.

6. Please clarify within the manuscript whether the model explicitly incorporates the following strategies: providing antiretroviral therapy (ART) for undiagnosed cases and cotrimoxazole for cases testing positive for HIV, as well as whether it considers the progression of the disease among children with HIV. This clarification is important for understanding the interventions considered in the study and their potential impact on disease progression and treatment outcomes.

7. Please clarify within the manuscript whether the model explicitly incorporates the following strategies among children with HIV: 

a) Providing antiretroviral therapy (ART) for undiagnosed cases and cotrimoxazole for cases testing positive for HIV

b) Disease progression among children with HIV into the mode

8. Could the author kindly include the confidence interval (CI), if possible, for the risk ratio of strategies in comparison to the current strategy?

9. Could the authors please provide a summary of the main findings? This summary should be provided as a paragraph at the beginning of the discussion section.

10. Could the authors please provide a clear and detailed discussion of the limitations of the research? Additionally, it would be valuable to explore the need for further research, such as conducting a cost-effectiveness analysis and examining the potential impact on antibiotic resistance for the proposed strategies.

Reviewer #4: The manuscript reports the impact of different cotrimoxazole strategies on mortality among children born to HIV-positive mothers. This work may help inform WHO recommendations on cotrimoxazole prophylaxis given some concerns on its necessity among HIV-exposed children given concerns on antibiotic resistance. 

The manuscript is well written and informative: I only have minor comments regarding the methods:

Model structure

In my opinion, the author should clearly state what they consider as "early initiation diagnosis (EID)" before detailing the model in order to clarify it. 

Two model structures are provided: Figure 1 and Supplementary table 1. In my opinion, providing two model structures may be misleading and authors may consider clearly explaining the differences between both. Regarding the model structure in Supplementary Material, the document states that infants are categorized according to whether they have acquired perinatal HIV and whether they undergo early infant diagnosis. If I understand well, perinatal HIV is defined as acquiring HIV by 6 weeks, but the model classifies the children as born with HIV or not, which in my opinion is different. 

Data sources and parameters

Data on subsequent testing after EID and engagement of children with HIV services: in my opinion it would be important to support the assumptions with literature. For Mozambique, I suggest to read the "Inquérito de Indicadores de Imunização, Malária e HIV/SIDA" (https://www.misau.gov.mz/index.php/inqueritos-de-saude?download=566:relatorio-final-imasida). Despite it dates from 2015, I think it might provide useful information to inform the assumptions.

[LINK]

---

## [Decision Letter · Decision Letter 2]

24 Oct 2023

Dear Dr. Mathur,

Thank you very much for submitting your manuscript "Estimating the Impact of Alternative Programmatic Cotrimoxazole Strategies on Mortality Among Children Born to Mothers with HIV: A Modelling Study" (PMEDICINE-D-23-01623R2) for consideration at PLOS Medicine. 

Your paper was re-evaluated by three independent reviewers, including a statistical reviewer, and discussed among all the editors here. The reviews are appended at the bottom of this email and any accompanying reviewer attachments can be seen via the link below:

[LINK]

In light of these reviews, I am afraid that we will not be able to accept the manuscript for publication in the journal in its current form, but we would like to consider a revised version that fully addresses Reviewer 1's comments. Obviously we cannot make any decision about publication until we have seen the revised manuscript and your response, and we plan to seek re-review by one or more of the reviewers. 

We expect to receive your revised manuscript by Nov 14 2023 11:59PM. Please email me (lgaynor@plos.org) if you have any questions or concerns.

We look forward to receiving your revised manuscript. 

Sincerely,

Louise Gaynor-Brook, MBBS PhD

plosmedicine.org

lgaynor@plos.org

In your revised submission, please ensure that you fully address the comments from Reviewer 1, relating to strengthening the model to take into account cotrimoxazole usage among infants not engaged in care, and consideration of the Mpepu RCT as suggested.

Comments from the reviewers:

Reviewer #1: Mathur and colleagues have significantly improved their manuscript entitled "The Impact of Alternative Programmatic Cotrimoxazole Strategies on Mortality Among Children Born to Mothers with HIV: A Modelling Study" by addressing opportunities raised by the editorial staff of PLOS Medicine, as well as that of the reviewers. Again noted is the considerable effort in further customizing the multiple strategies to quantify predicted mortality. There remain opportunities to further strengthen the manuscript, including:

The premise of this modeling exercise is that infants and children living with HIV are not diagnosed and treated proximal to the incident HIV infection. While cotrimoxazole does have mortality benefit among children infected with HIV, all models presume there will be no change in pediatric HIV testing and rapid antiretroviral treatment initiation. In the limitations paragraph of the discussion section it should be pointed out that investment in ensuring that all children HIV-exposed uninfected receive cotrimoxazole until risk of HIV infection has been eliminated may not represent the optimal health investment compared to investing in timely testing and treatment, an alternative known to substantially reduce morbidity and mortality, preserve immune function, and limit seeding of the HIV viral reservoir, that latter two of which cotrimoxazole cannot affect.

At lines 222 through 224, while it is appropriate for the purposes of modeling to assume that children who do not test at 6 weeks would have a lower prevalence of testing at 9 months and following cessation of breastfeeding, as the authors appropriately note this group is less likely to be engaged in care, it is imperative that this population also have a much lower use of cotrimoxazole in model. If they are not engaged in care, they will not be prescribed cotrimoxazole. Reducing the proportion of infants in this group who actually take cotrimoxazole will likely substantially reduce the overall mortality benefit of a policy of universal use of cotrimoxazole in high burden HIV settings.

At lines 258 and 259, rationale is used for selection of conservative estimates of mortality benefit with cotrimoxazole use, since the CHAP study included very few infants. Why wouldn't the authors use the Mpepu study, a more current study were infants HIV-exposed uninfected were randomized to cotrimoxazole or placebo through 15 months of life? This is particularly important as at lines 372 through 374 it is stated that there is no randomized controlled trial data for cotrimoxazole among breastfed infants. The Mpepu study in Botswana was randomized controlled trial that included breastfed infants.

At lines 361 through 362, in addition to recognizing to the need to focus on reducing vertical transmission, it would be equally important to highlight, where vertical transmission does occur, timely testing and initiation of antiretroviral treatment.

Under "Why was this study done?", under the first bullet, how would one ascertain the period of HIV infection risk through breastfeeding has ended without performing testing? It might be clearer to the reader to state "until the child is no longer at risk for HIV acquisition and documentation of a negative HIV test appropriately timed after the risk has concluded is available".

For the statement between lines 91-94, it would be important to use the term "model" in place of "determine" and/or to state "would be predicted to increase mortality" rather than "would increase mortality".

At line 100, can you clarify how the model "incorporates the HIV status of the infant"? All infants/children in the model start off HIV-exposed uninfected and that same bullet point goes on to explain that perinatal and postnatal transmission rates are employed in the model.

Please consider changing the text at lines 121 and 122 to read "Policymakers need to weigh the risks and benefits of guidelines for cotrimoxazole prophylaxis among infants HIV-exposed uninfected, recognizing that vertical transmission rates and timely test and treat programming significantly influences predicted mortality outcomes".

The text between lines 164 and 166 indicate that modeling of mortality impact of alternative cotrimoxazole strategies in four high-burden settings was performed to inform potential cotrimoxazole recommendations in different contexts. However, modeling in four high-burden settings really does not afford the ability to inform recommendations in different contexts. It is more accurate to simply state "Here, we model the mortality impact of alternative cotrimoxazole strategies in four high-burden settings".

The manuscript has many places where first person language is not presented, including at line 92 (HIV-exposed infants should be infants HIV-exposed), line 173, line 204, line 246, and line 392.

PMTCT is used for the first time at line 115. Please spell it out.

Lines 180 and 181 do not need "Early Infant Diagnosis" as the abbreviation has already been used previously.

At line 87, the "now" is not necessary.

Reviewer #2: The authors have done a good job of replying to all my questions and concerns. I have no further concerns. 

Reviewer #3: The authors have diligently addressed all comments, and I now find the manuscript suitable for publication in PLOS Medicine. The only minor revision I identified is related to the "What Did the Researchers Do and Find?" section. Specifically, there is duplication of the phrase "limited by" that should be removed. The sentence should read as follows: "Our study is limited by the lack of a cost-effectiveness analysis, data on cotrimoxazole uptake, and comprehensive antimicrobial resistance surveillance data in sub-Saharan Africa."

[LINK]

---

## [Decision Letter · Decision Letter 3]

11 Dec 2023

Dear Dr. Mathur,

Thank you very much for re-submitting your manuscript "Estimating the Impact of Alternative Programmatic Cotrimoxazole Strategies on Mortality Among Children Born to Mothers with HIV: A Modelling Study" (PMEDICINE-D-23-01623R3) for review by PLOS Medicine.

I have discussed the paper with my colleagues and the academic editor and it was also seen again by two reviewers. I am pleased to say that provided the remaining editorial and production issues are dealt with we are planning to accept the paper for publication in the journal.

[LINK]

We expect to receive your revised manuscript within 1 week. Please email me (lgaynor@plos.org) if you have any questions or concerns.

We look forward to receiving the revised manuscript by Dec 18 2023 11:59PM.   

Sincerely,

Louise Gaynor-Brook, MBBS PhD

lgaynor@plos.org

plosmedicine.org

Thank you for your responses to the comments from the editors and reviewers in your revised submission. A few minor issues detailed below require further attention before we are able to accept your manuscript for publication. 

Comments from the Academic Editor:

In some places in the discussion, the authors could change the order of the relevant sentence to start with insisting that HIV diagnosis and immediate treatment of infants found to be infected is priority, and that provision of cotrimoxazole should not be seen as a 'way out' and reducing the costs of the HIV prevention and care programmes, before stating that co-trimoxazole should be provided to HEU. 

RE: availability of trials in breastfed populations - the authors should amend their sentence re no trials being available to note that only limited data is available re use of CTX in HEU in breastfeeding populations, and reference the Mpepu trial, where 20% of infants were breastfed. In many high HIV prevalence settings breastfeeding does happen but often not for that long - for a variety of reasons - which likely underlies the finding in Arikiwa that breastfeeding was not clearly associated with mortality in HEU infants.

There is no need to compare mortality between Mpepu and CHAPS trials.

Requests from the Editors:

To help us extend the reach of your research, please provide any Twitter handle(s) that would be appropriate to tag, including your own, your coauthors’, your institution, funder, etc. 

Author Summary:

Line 111 - please revise to ‘Increased predicted morality…’ and ‘although it was predicted that cotrimoxazole would have fewer benefits’ or similar

Methods:

Please state early in the Methods section that your study did not have a prospective protocol. 

Please add the following statement, or similar, to the Methods: "This study is reported as per the Transparent reporting of a multivariable prediction model for individual prognosis or diagnosis (TRIPOD) guideline (S1 Checklist)."

Results: 

Lines 285, 295, 304, 309 - please revise to ‘predicted excess mortality’ (or similar)

Line 320 - please specify which supplementary files are relevant to each setting.

Figures:

Please ensure that all abbreviations are defined in the figure legend of each figure, including those in the supplementary information.

Comments from Reviewers:

Reviewer #1: Mathur and colleagues have made further changes to their manuscript entitled "The Impact of Alternative Programmatic Cotrimoxazole Strategies on Mortality Among Children Born to Mothers with HIV: A Modelling Study" in an effort to be responsive to reviewer suggestions. Two key points which were not adequately addressed in this latest revision should be addressed prior to publication. 

First, it is again requested that in the limitations paragraph of the discussion section it be pointed out that investment in ensuring that all children HIV-exposed uninfected receive cotrimoxazole until risk of HIV infection has been eliminated may not represent the optimal health investment compared to investing in timely testing and treatment, an alternative known to substantially reduce morbidity and mortality, preserve immune function, and limit seeding of the HIV viral reservoir, the latter two of which cotrimoxazole cannot affect. While the response from the authors to this previous request characterize prescribing of cotrimoxazole as a "safety net" when gaps in the test-to-treat cascade exist, the "safety net" is the health care system. The modeling exercise does not sufficiently account for lack of engagement in the health care "safety net". It is again requested that the wording be clear that funding and programming of cotrimoxazole may not represent the best investment compared to investment in the test-to-treat cascade. It is recommended that this appear in the limitations paragraph of the discussion section.

Secondly, although the authors do reference the Botswana-based Mpepu study and the South African study, lines 379 through 383 continue to state:

"There are no randomized controlled trial data for cotrimoxazole among breastfed infants, but it is unlikely that mortality benefits are substantially lower than in the original CHAP trial, given the more rapid disease progression at this age, lack of prognostic markers, and higher risk of Pneumocystis jirovecii."

As previously pointed out, the Mpepu study does represent a randomized controlled trial where 20% of the infants in the study were breastfeeding and these infants were randomized to cotrimoxazole or placebo. Please change the statement in the discussion to reflect this and consider comparing Mpepu mortality data to that of the original CHAP trial. If the fact that mortality is only reported through 18 months is the reason Mpepu mortality data was not used, then the text should reflect that rather than indicating that no randomized clinical trial has occurred.

Reviewer #3: To enhance the clarity and precision of the manuscript, I propose a revision to lines 410 and 411 in the Discussion section as detailed below:

Lastly, this study did not consider the economic implications of the proposed strategies. Future research should investigate the potential cost savings and the financial impacts of implementing each strategy on national budgets.

[LINK]

---

## [Editor Report · Decision Letter 4]

10 Jan 2024

Dear Dr Mathur, 

On behalf of my colleagues and the Academic Editor, Prof. Marie-Louise Newell, I am pleased to inform you that we have agreed to publish your manuscript "Estimating the Impact of Alternative Programmatic Cotrimoxazole Strategies on Mortality Among Children Born to Mothers with HIV: A Modelling Study" (PMEDICINE-D-23-01623R4) in PLOS Medicine.

PRESS

Sincerely, 

Louise Gaynor-Brook, MBBS PhD 

Senior Editor 

PLOS Medicine